# The Effects of Calorie Restriction on Autophagy: Role on Aging Intervention

**DOI:** 10.3390/nu11122923

**Published:** 2019-12-02

**Authors:** Ki Wung Chung, Hae Young Chung

**Affiliations:** 1College of Pharmacy, Kyungsung University, Busan 48434, Korea; 2College of Pharmacy, Pusan National University, Busan 462414, Korea

**Keywords:** aging, autophagy, calorie restriction (CR), CR mimetic

## Abstract

Autophagy is an important housekeeping process that maintains a proper cellular homeostasis under normal physiologic and/or pathologic conditions. It is responsible for the disposal and recycling of metabolic macromolecules and damaged organelles through broad lysosomal degradation processes. Under stress conditions, including nutrient deficiency, autophagy is substantially activated to maintain proper cell function and promote cell survival. Altered autophagy processes have been reported in various aging studies, and a dysregulated autophagy is associated with various age-associated diseases. Calorie restriction (CR) is regarded as the gold standard for many aging intervention methods. Although it is clear that CR has diverse effects in counteracting aging process, the exact mechanisms by which it modulates those processes are still controversial. Recent advances in CR research have suggested that the activation of autophagy is linked to the observed beneficial anti-aging effects. Evidence showed that CR induced a robust autophagy response in various metabolic tissues, and that the inhibition of autophagy attenuated the anti-aging effects of CR. The mechanisms by which CR modulates the complex process of autophagy have been investigated in depth. In this review, several major advances related to CR’s anti-aging mechanisms and anti-aging mimetics will be discussed, focusing on the modification of the autophagy response.

## 1. Introduction

### 1.1. The Autophagy Process

Autophagy is an evolutionarily well-conserved process that occurs in all eukaryotic cells from yeast to human [1]. The highly complex autophagy-related signaling pathways have been extensively studied for the last 30 years, and they have been elucidated through the combined study of genetics and physiology in various species [2]. At least three different forms of autophagy have been identified so far: macro-autophagy, micro-autophagy, and chaperone-mediated autophagy. All three forms depend on lysosomal degradation, with macro-autophagy (hereafter referred to as autophagy) being the most prevalent form. Once activated, autophagy involves the sequestering of cytosolic components (damaged cell organelles, proteins, or other macromolecule nutrients) by phagophores that mature into autophagosomes, which are double membrane vesicles [2,3]. Autophagosomes further translocate and fuse to the acidic lysosome and form the autolysosome, where degradation and recycling occur. The diverse substrates and basal activity of these processes suggest that cells are highly dependent on it for maintaining cellular homeostasis. The importance of maintaining an adequate autophagy response has been demonstrated under both physiologic and pathologic conditions [4].

### 1.2. Molecular Machinery of the Autophagy Process

The molecular mechanisms and signaling pathways controlling autophagy have been extensively studied [5]. Autophagy begins with the *de novo* production of autophagosome components, followed by assembly driven by the concerted action of a group of proteins named ATG (autophagy-related genes). As the detailed molecular machinery of the autophagy process has been previously described in several review articles, only its overall features will be discussed in this review. At the start of the autophagy process, phagophore formation is initiated from the endoplasmic reticulum (ER)–mitochondrial interface, and further elongation of the phagophore depends on the Golgi and plasma membranes. The progression of autophagosome formation is largely characterized by the recruitment of ATG proteins to the phagophore [6].

The formation of the UMC-51-like kinase 1 (ULK1, homologous to yeast ATG1) complex is the earliest event in the formation of the autophagosome. ULK1 activation lies upstream of other ATG protein recruitment, and ULK1 kinase activity is required for the recruitment of the VPS34 complex (a class III PI3-kinase) to the phagophore. This is crucial for the phosphorylation of phosphatidyl inositol (PtdIns) and the subsequent production of PtdIns 3-phosphate. The further recruitment of phospholipid-binding proteins to the phagophore is important for the stabilization of protein complexes near the autophagosome formation site. Two conjugation systems are involved in the vesicle elongation process. The conjugation of ATG5 to the ATG12 complex requires the ubiquitin-like conjugation system involving ATG7 and ATG10. The conjugated ATG5–ATG12 complex is needed to further conjugate phosphoethanolamine (PE) to ATG8 (microtubule-associated protein 1 light chain 3; LC3). ATG4, ATG7, and ATG3 are required for this conjugation process. The conversion of LC3 from LC3-I (soluble form) to LC3-II (vesicle associated form) by PE conjugation is thought to be required for the closure of the expanding autophagosomal membrane. Finally, the matured autophagosome is fused with the lysosome to fulfill the main purpose of the process, culminating with the degradation and recycling of substrates in the autophagosome.

### 1.3. Autophagy Is Regulated by Nutrient-Sensing Signaling

A variety of physiologically important stimuli induce the autophagy process, including organelle (ER, mitochondria) damage, hypoxia, and inflammation [2]. However, nutrients and energy stress are the most powerful regulators of the autophagy process [7]. Changes in the cellular energy status such as the withdrawal of nutrients, such as glucose and amino acids, induce the activation of the autophagy process, from initiation to termination [8]. Nutrient levels can be directly recognized by the upstream signaling machinery of autophagy to regulate its initiation in response to the changing cellular energy levels (Figure 1).

Of all the nutrient-associated signaling molecules, mammalian target of rapamycin (mTOR) has been shown as one of the key upstream modulators of autophagy signaling [9,10]. mTOR is a highly conserved serine/threonine kinase that is regulated by multiple signals including energy levels, growth factors, and other cellular stressors, to coordinate cell proliferation/growth and maintain energy homeostasis. mTOR forms a complex, which is known as mTORC1 (mTOR complex 1) and mTORC2 (mTOR complex 2). mTORC1 is related to autophagy signaling changes and is activated in the presence of nutrients or growth factors. mTORC1 is usually activated under nutrient-rich conditions [11]. It can be directly activated by an increased concentration of amino acids in the cell or as downstream signaling through the action of growth factors [11,12]. Once activated, mTORC1 directly phosphorylates ULK1 [13]. Critically, the activation of mTORC1 is sufficient to inhibit autophagy in the presence of sufficient nutrients [14]. The direct repression of ULK1 kinase by mTORC1 is also well conserved across species [15]. Other components of the ATG complex directly interact with mTORC1 and repress the autophagy process [16]. Furthermore, mTORC1 can indirectly suppress autophagy by controlling lysosome biogenesis [17,18]. The transcription factor EB (TFEB) is responsible for the transcription of lysosomal and autophagy-related genes [19]. mTORC1-mediated TFEB phosphorylation decreases its transcriptional activity, thus decreasing the overall expression of autophagy-related gene expression [20,21].

Under nutrient-deficient conditions, the activation of autophagy is regulated by several well-known nutrient-sensing signaling proteins. One of the most prominent players of nutrient deprivation sensing is the AMP-activated protein kinase (AMPK) [13,22]. The molecular ratio of ATP to AMP reflects the cell’s energy levels, and increased levels of AMP represent an internal cell warning system that induces the cell to save energy for the maintenance of metabolic homeostasis. AMP is directly sensed by AMPK, and activated AMPK has been characterized and shown to have multiple functions in the regulation of cellular metabolism. There are several mechanisms by which AMPK induces autophagy. First, AMPK directly phosphorylates ULK, which is a process that is required for ULK1 activation and the initiation of autophagy under nutrient deprivation conditions [13]. The interaction between AMPK and ULK1 can be blocked by mTORC1-mediated ULK1 phosphorylation, indicating an intricate connection between these two pathways. Secondly, AMPK is a negative regulator of the mTOR signaling pathway [23]. Mechanistically, AMPK directly phosphorylates the tuberculosis-associated complex (TSC), which is a negative regulator of mTORC1 activation. AMPK also directly phosphorylates the Raptor subunit of the mTORC1 complex, increasing the degradation of the mTORC1 complex. These studies clearly demonstrated that AMPK, a critical regulator of nutrient availability, is able to regulate autophagy activity by coordinating mTOR-dependent and independent mechanisms.

### 1.4. Autophagy and Aging

Aging is associated with various changes including genomic instability, loss of proteostasis, epigenetic alterations, and deregulated nutrient-sensing pathways [24]. These changes are also associated with numerous age-related diseases including cardiovascular diseases, neurodegenerative diseases, and metabolic diseases. Among the changes that occur during aging, some are associated with autophagy-related signaling pathways [24,25]. A decline in the overall proteolytic activity and an altered nutrient-sensing signaling are directly associated with autophagy. Indeed, decreased autophagy with aging has been reported extensively in a broad range of organisms, where a progressive accumulation of damaged proteins and cellular organelles was shown to occur [26]. The decreased level of autophagy-related gene transcripts and proteins has been detected in nematodes and the fruit fly [27,28,29]. Aged tissues from mammals and humans also showed a lower expression of key autophagy proteins [30,31,32]. Consistent with changes in the levels of autophagy components, recent studies further showed a decreased overall autophagic capacity during aging in *C. elegans* [33]. Electron microscopy observations showed an age-related accumulation of autophagic vacuoles, which represents the blockage of autophagy flux. Similarly, the overall proteolysis activity is impaired during the aging process, and long-lived proteins that were not properly degraded have been detected in the liver in aged rats [34].

Further striking evidence between aging and autophagy comes from several genetic models of impaired autophagy. An unbiased screen for aging factors in yeast, nematodes, and fruit fly revealed short-lived mutants with defects in autophagy [27,35,36]. Moreover, in knockout mice, whole body deletion of autophagy-related genes led to early postnatal death, indicating an essential role of autophagy in the overall maintenance of physiological processes [37,38,39]. Tissue-specific conditional knockout mice models also revealed the multiple phenotypes of aging, including the aggregation and accumulation of intracellular proteins, cellular organelles, and other macromolecules [40,41,42,43]. The loss of autophagic activity in these models is likely to increasingly constrain the ability of the cells to maintain quality control, leading to the accumulation of toxic insults, and resulting in aging and age-associated pathologies [3]. On the other hand, accumulating evidence suggests that experimentally enhanced autophagy extends the lifespan and delays the aged phenotype. The overexpression of specific autophagy genes can extend the lifespan in several species. The upregulation of autophagic activity can extend longevity in *C. elegans*, as well as in the yeast, while the ubiquitous overexpression of Atg5 in mice is sufficient to stimulate autophagy and extend the lifespan [44,45]. Collectively, these observations indicate that changes in autophagic activity may be associated with longevity and that augmenting autophagic function may be an effective approach to delay aging and promote longevity in different species, including in mammals.

The mechanisms by which autophagy components or autophagic processes decrease with age remain unclear. Since the autophagy process, from initiation to completion, is complex and associated with various steps and different proteins, it is likely that the mechanisms contributing to age-associated autophagy decrease are multifactorial. The most plausible regulatory mechanism contributing to suppressed autophagy in aging is a change in the upstream signaling during autophagy initiation. Two important nutrient-sensing proteins, mTOR and AMPK, play an important role in the regulation of the initiation of autophagy [10,13]. Furthermore, these factors reflect the status of a cell, such as hormonal regulation (outside the cell) and nutrition stress (inside the cell). The nutrient sensor mTOR strongly inhibits not only the initiation of autophagy but also exerts an inhibitory effect on multiple steps in the autophagy process. It is possible that increased mTOR signaling during aging plays an important role in the age-associated suppression of autophagy. Since increased mTOR activity has been reported in various age-related diseases including metabolic and degenerative disorders, it is plausible that increased mTOR signaling is the predominant cause for the downregulation of the overall autophagy process [46]. Unlike mTOR, which is usually hyperactivated during aging, the activity or expression of AMPK is typically suppressed [47]. It is plausible that decreased AMPK might influence or suppress autophagy and act in concert with mTOR. To this end, although mechanisms disrupting autophagy signaling during aging are multifactorial, it is clear that modifications in its upstream pathways are critical for its regulation.

Another possible mechanism responsible for the decreased autophagy observed in aging is transcriptional regulation. TFEB has been previously described as a regulator of autophagy-related gene transcription; however, recent studies have revealed other important transcription factors that regulate the gene expression of autophagy-related proteins. The fasting transcriptional factor CRE-binding protein (CREB) is upregulated by glucagon under nutrient deprivation conditions, and it also upregulates autophagy gene expression including ATG7, ULK1, and TFEB. In addition to CREB, peroxisome proliferation factor-activated receptor α (PPARα), another transcription factor playing a role in starvation, also directs the transcription of autophagy genes [19,48,49]. Both transcription factors may act in concert to increase autophagy-related gene expression. The genetic deletion of both transcription factors reduced autophagy and led to an inadequate metabolic response, particularly under nutrient deprivation. Although there is no direct evidence of whether they play a role in defective autophagy during aging, there is some evidence that they are important and dysregulated during aging [50,51,52]. Further studies will be necessary to reveal the relationship between these transcription factors and defective autophagy during aging.

## 2. Calorie Restriction (CR) Modulates Autophagy Processes

### 2.1. Introduction to Calorie Restriction

Calorie restriction (CR) has been shown to be an established life-extension method regulating age-related diseases as well as aging itself. Although different in methodology (usually 20%–40% ad libitum intake, 40% reduction in most cases), CR showed a prolonged lifespan in a wide range of species from yeast to non-human primates, and supports healthy human aging [53]. Furthermore, CR exerts preventive effects on various age-related conditions such as cancer, neurodegenerative diseases, cardiovascular, and other metabolic diseases [54]. The diverse efficacy of CR in counteracting aging and age-related diseases has made it the golden standard of aging intervention studies. Although the anti-aging effects of CR are reproducible, the exact mechanisms of how CR exerts its anti-aging effects are debatable, because CR regulates several different aspects of physiology. These changes include modifications in the energy-sensing signaling, oxidative stress, inflammation, and other intercellular and intracellular processes. Among the many changes induced by CR, energy production and utilization is the most directly regulated signaling exerted by CR [55,56]. Since reduced energy intake and changes in nutritional status following CR may change the molecular signaling pathways associated with energy-sensing mechanisms, other mechanisms may be secondary effects to this process.

### 2.2. Evidence for the Beneficial Effects of CR-Mediated Autophagy

Based on the induction mechanism of autophagy and its role during starvation, it was predicted that CR might induce the autophagic process. Indeed, under many different settings of nutrient deprivation conditions, including in CR, autophagy is induced to regulate the organism’s homeostasis. Although it is clear that CR represents a strong physiologically autophagic inducer, it is uncertain whether autophagy contributes to the anti-aging effects of CR. Recently, several studies have shown that autophagy induction was essential for the anti-aging effects of CR (Table 1). CR was shown to promote longevity or protect from hypoxia through a Sirtuin-1-dependent autophagy induction process [57,58]. Another study also showed that life extension through methionine restriction required autophagy activation [59]. Growing evidence supports the notion that autophagy has a substantial role in the beneficial effects of CR [60,61]. In addition to research on longevity, other studies have shown that CR robustly induces autophagy under various physiological and pathological conditions, and that it has a protective effect in the maintenance of normal functions in the organism. In the following section, the protective role of autophagy under CR conditions will be discussed.

## 3. Protective Effects of CR-Induced Autophagy on Different Organs

The substrates of autophagy include important macronutrients such as glycogen and lipid droplets [72,73]. Under CR conditions, it is essential for cells to use their internal nutrient stores. The breakdown products derived from autophagy provide substrates for biosynthesis and energy generation. The redistribution of nutrients, under starved or CR conditions, is essential for the cells to adapt to the changed nutritional environment. Indeed, metabolic tissues show the most dramatic changes in autophagy regulation under nutrient-starved conditions, suggesting its important role in the regulation of metabolism.

### 3.1. Liver

The importance and original concept of autophagy was first described in the liver, where high levels of enzymes and cellular organelles associated with lysosomal degradation are found. Liver autophagy plays an important role under physiologic and pathologic conditions by contributing to the recycling of organelles, as well as macronutrients [74]. Recent evidence showed that the role of liver autophagy under normal physiological conditions is to regulate the nutrient degradation systems, such as glycogenolysis and lipid droplet degradation [42,75]. Furthermore, lipid droplet degradation in hepatocytes (lipophagy) is particularly important under pathologic conditions such as in non-alcoholic fatty liver disease, steatohepatitis, and in hepatocellular carcinoma [75,76,77]. The deficient autophagic response aggravated not only lipid accumulation but also other pathologic features of liver disease. Liver autophagy is also impaired during aging. The base level of autophagy as well as autophagy induced by stress responses is impaired in the aged liver, making it vulnerable to liver damage [72].

The effects of CR on liver autophagy were assessed in several studies. Wohlgemuth et al. evaluated the effects of life-long CR in Fisher rats [62]. They found that life-long CR did not cause a substantial change in the expression of autophagic proteins in the liver. However, other studies using different CR settings found different results. Donati et al. assessed the effect of CR following alternate day fasting [63]. When studying the rate of autophagic proteolysis in the isolated livers, they found that maximum rates of autophagy were achieved in the CR groups compared to controls. A more recent study by Luevano-Martinez et al. showed the effect of CR on the induction of autophagy in liver mitochondria [64]. They isolated mitochondria from the livers of controls, and after 4 months of a CR schedule, they found an increase in the LC3-II/LC3-I ratio in CR livers, indicating enhanced liver mitochondrial autophagy. Derous et al. found similar results when they evaluated the effect of graded levels of CR on autophagy using the hepatic transcriptome [65]. Mice were subjected to a graded level of CR (from 0% to 40% CR) for 3 months, following which a significant increase in autophagy levels was observed that correlated with increased levels of CR. In the liver, the autophagy response is generally increased following CR, independently of the method used to induce CR.

In addition to CR, fasting also induced a robust hepatic autophagy. Although fasting is different from a consistent pattern of CR, they share some common features. Researchers have identified that a fasting-induced autophagy response is a fundamental process during food deprivation and is an important protective response in the regulation of metabolism [78,79].

### 3.2. Muscle

The skeletal muscle is the most abundant body tissue (comprising approximately 40% of the body weight) and is a dynamic tissue consistently adapting to metabolic demands. To meet the high metabolic demand, autophagy proteolytic systems engage in metabolic regulation [80]. In the muscle, autophagy regulates protein degradation and provides amino acids for energy production [43,81]. This is particularly important under nutrient-deprived or stress conditions to maintain adequate energy production. Recent studies have shown that basal autophagy is crucial for the maintenance of muscle physiology, and that a maladaptive autophagy is implicated in various muscle diseases, including muscular dystrophy, sarcopenia, and myofibril degeneration [31,66,82,83,84].

Several studies have shown the ability of CR to induce muscle autophagy and its beneficial effects. Wohlgemuth et al. investigated the effects of aging and mild CR on skeletal muscle autophagy and lysosome-related proteins [66]. They found LC3-I and LAMP-2 accumulation, suggesting an age-related decline in autophagic degradation. Age-related changes were inhibited by CR, concluding that mild CR attenuated the age-related impairment of autophagy in skeletal muscle in rodents. More evidence comes from a recent clinical trial study. Yang et al. showed that long-term CR enhanced the overall quality-control processes in human skeletal muscle [67]. They found that several autophagy genes, including ULK1, ATG101, beclin-1, LC3 were significantly upregulated in response to CR. Furthermore, they found decreased muscle inflammation, suggesting another beneficial role of CR on muscle biology. The study by Gutierrez-Casado et al. also showed a prominent effect of CR on autophagy in the muscle [68]. CR resulted in decreased levels of p62, suggesting a possible increase in autophagy flux. Although not experimentally demonstrated, Lee et al. suggested the importance of the role of autophagy on muscle stem cell regeneration induced by CR [85]. CR not only improved stem cell regenerative capacity but also enhanced the engraftment capacity of muscle stem cells [86]. CR-induced autophagy may prime the improvement in oxidative stress and increase mitochondrial activity in muscle stem cells, contributing to their beneficial regenerative effects in muscle.

### 3.3. Adipose Tissue

Adipose tissue is another important metabolic tissue that plays an important role in lipid storage during energy-sufficient conditions. A reduction in adiposity is the hallmark of CR, which is a consequence that may result from hormonal changes [87]. Although it is clear that autophagy induces lipid degradation through lipophagy in the liver, the role of autophagy in the regulation of adipose tissue lipids is more complex [88]. Singh et al. first showed that adipose tissue autophagy regulates adipose tissue mass and differentiation [89]. They found that the knockdown of Atg7, an essential autophagy gene, inhibited lipid accumulation and decreased the protein level of several adipocyte differentiation factors. Furthermore, they demonstrated that the adipocyte-specific Atg7 knockout mouse had a lean phenotype with decreased white adipose mass and enhanced insulin sensitivity. However, more recently, Cai et al. showed a protective effect of autophagy in mature adipocyte function [90]. They showed that autophagy proteins are required for adequate mitochondrial function and that the post-development ablation of autophagy caused insulin resistance.

The defective regulation of adipose tissue autophagy has been detected in mice and human obesity [69]. In mice models of obesity and in obese humans, autophagy-related genes and proteins were found to be significantly upregulated [91,92]. Although these results were interpreted as increased autophagy, at first, Soussi et al. showed that the autophagy flux was impaired in obesity [93]. This result was consistent with the conclusion derived from the work of Cai et al., showing that autophagy may play a protective role after maturation. Based on these results, it is clear that the maintenance of an appropriate activation of autophagy is needed in the adipose tissue. However, the role of CR in adipose tissue function is yet to be clarified. Nunez et al. showed that CR successfully increased autophagy in lean mice, but in obese mice, autophagy induction did not occur, suggesting that similarly to previous reports, the autophagic response is defective during obesity [69]. Ghosh et al. studied the effects of aging and CR on adipose tissue autophagy and found a diminished autophagy activity with aging, contributing to aberrant ER stress and inflammation in aged adipose tissue [70]. They also showed that autophagy activity was enhanced in the CR mice with a concomitant decrease in ER stress and inflammation. Taken together, CR has beneficial effects on adipose tissue, at least partly through the induction of the autophagy response.

### 3.4. Kidney

Kidneys also show beneficial effects from CR, including the induction of autophagy. In the canonical concept of metabolism, the kidney is not an active participant. However, the kidney can participate and play an important role in the metabolism of carbohydrates, proteins, and lipids [94,95]. Renal tubule cells have a high basal level of energy consumption and depend on the β-oxidation of fatty acids to generate adequate amounts of ATP [96]. Furthermore, proximal tubule cells generate glucose through gluconeogenesis, especially under nutrient-deficient conditions, and contribute to the total blood glucose level [97]. For these reasons, suitable autophagy is important for the maintenance of normal kidney physiology by regulating adequate metabolic processes and organelle quality. Defects in autophagy have been found to worsen conditions in several types of kidney diseases [98,99]. CR is known to have beneficial effects in the kidney both under physiologic and pathologic conditions [100]. In addition, CR also leads to a delayed age-associated kidney dysfunction and to structural changes [50]. Among several suggested mechanisms that explain the beneficial effects of CR in the kidneys, increased autophagy activity is an important one.

Kume et al. designed a 12-month-long CR schedule in 12-month-old mice to assess the effect of aging and CR on autophagy [58]. In comparison to the control group, CR resulted in healthy mitochondria with numerous autophagosomes in the kidney. In addition, a lower level of p62 was found in the kidney of the CR mice. The ratio of LC3 conversion and LC3 puncta were higher in the CR mice, indicating that CR-mediated autophagy increased mitochondrial integrity and protected from age-associated kidney damage. Ning et al. showed a similar result using a short-term calorie restriction model [71]. CR groups had a 40% calorie restriction for 8 weeks, and showed increased autophagy flux, autophagy-related gene expression, and reduced oxidative damage. CR also significantly decreased p62 expression and polyubiquitin aggregates.

Chung et al. also showed that short-term CR reduced age-associated renal fibrosis [50]. They found that reduced PPARα expression during aging impaired lipid metabolism and induced interstitial fibrosis in the kidney. PPARα knockout mice showed an early onset of age-associated kidney fibrosis. Although they only focused on lipid metabolism for the regulatory role of PPARα and did not check for autophagy changes in their model, PPARα plays an important role in the expression of autophagy-related genes; therefore, it is plausible that autophagy might have played a role in mediating the anti-fibrosis effects of CR in their model. Collectively, these studies strongly suggest that CR effectively induces autophagy in aging and diabetic mice and plays a protective role in these settings.

## 4. Benefits of Intermeal Fasting in Autophagy: Is CR the Only Solution?

Recently, an interesting study by Martinez-Lopez et al. demonstrated a pivotal role for autophagy under nutritional conditions other than CR [78]. They introduced an isocaloric twice-a-day (ITAD) feeding model with the same amount of food consumption in total as ad libitum controls. These mice were exposed to food at two short time intervals, early and late in the diurnal cycle. The concept of this model is different from calorie restriction because the total food intake is the same as the controls. ITAD still leads to intermeal fasting, which induces various physiological changes, including the autophagy process. ITAD feeding impacted autophagy flux in multiple organs including liver, adipose tissue, muscle, and neurons. ITAD feeding promoted multiple metabolic benefits in organs where autophagy was increased, and further experiments demonstrated a tissue-specific contribution of autophagy to the metabolic benefits of ITAD feeding by use of tissue-specific autophagy knockout models. Finally, in an aging and obesity model, it was concluded that consuming two meals a day without CR could prevent metabolic syndrome through the activation of autophagy. This study could easily translate to humans, as ITAD is more feasibly applied than CR. If a similar regimen was applied in humans, it could provide some beneficial effects such as autophagy induction and ultimately prevent various age-associated metabolic diseases.

More recently, Stekovic et al. showed a prominent effect of alternate day (AD) fasting on aging in non-obese humans [101]. AD fasting significantly improved physiological and molecular markers; it also improved cardiovascular markers with reduced fat mass and without any of the typical adverse effects. This study also emphasized that AD fasting can be tolerated more easily than continuous CR and lead to similar beneficial effects. Although they did not check whether the autophagic response played a role, it might be interesting to further investigate the effects of AD fasting on autophagy induction.

## 5. CR Mimetic as an Autophagy Inducer

CR could have beneficial effects that prolong human lifespan; however, it is challenging to implement, even in the case of short-term CR. Therefore, the development of drugs or compounds that mimic the effect of CR is an interesting topic of discussion among biologists and gerontologists [102]. Based on the pathways and proteins changed under CR conditions, many have started to investigate modulators that mimic the CR effect. Currently, several drugs and other compounds naturally occurring in the diet (nutraceuticals) have been shown to act as a CR mimetic through various mechanisms. The targets of mimetics include the glycolysis pathway, insulin/insulin-like growth factor signaling, mTOR, AMPK, sirtuins, and other pathways associated with CR. Interestingly, many well-known CR mimetics are directly or indirectly associated with autophagy regulation. The following discussion will focus on well-known CR mimetics that act through the regulation of autophagy (Figure 2).

### 5.1. Rapamycin, an mTOR Inhibitor

Rapamycin was initially described as an immune-suppressor drug and is a commonly referred compound for CR mimetics. In later studies, it has been demonstrated that rapamycin directly binds between FKBP12 and the mTOR kinase subunits of mTORC1, causing the inhibition of mTOR and its downstream signaling pathway [103]. The mTOR inhibitory activity of rapamycin gained attention because the activity and expression of mTOR is significantly increased in aging and in age-related diseases [104]. Furthermore, CR was shown to downregulate mTOR function, leading to an increased autophagy with decreased protein synthesis [105]. Rapamycin has been documented as delaying or ameliorating age-related diseases including metabolic diseases, cardiovascular diseases, Hutchinson–Gilford progeria syndrome premature aging phenotype, and neurodegenerative diseases [104,106]. Rapamycin also showed a lifespan extension effect in various animal models including in the yeast, fruit fly, and nematode [107]. In addition, the life-extension effect of rapamycin was also verified and replicated in mice by several independent groups [108,109].

Although rapamycin activates autophagy through the inhibition of mTOR, it also shows other beneficial effects through the regulation of other signaling pathways. mTORC1 is activated not only by nutrient levels in the cell but also by cell growth hormones. mTORC1 interacts with key proteins in the anabolic process such as S6K, 4E-BP1, and SREBP1c, and activates protein, lipid, nucleotide, and organelle synthesis such as mitochondria [104]. However, evidence has also demonstrated some side effects of rapamycin such as a suppressed immune system, increased incidence of diabetes, and nephrotoxicity [110]. The safety and side effects of rapamycin in the long-term use should be carefully considered.

### 5.2. Metformin, an AMPK Activator

Merformin is another interesting CR mimetic. It is a guanidine-based hypoglycemic agent that is used as a drug for the treatment of type-2 diabetes, and has the ability to increase insulin sensitivity through the activation of AMPK. Although it is commonly referred as an AMPK activator, it is unlikely that metformin directly binds to either AMPK or its activator LKB1 [111]. Evidence supports that metformin may increase AMPK activation by modulating ATP production in mitochondria [112]. Since AMPK is downregulated in many types of metabolic disease, metformin showed a particular beneficial effect in various age-related metabolic diseases [113]. Further studies have shown a lifespan extension effect of metformin. During the screening of CR mimetics, Dhahbi et al. first found that metformin treatment showed a similar transcriptional profile to that of CR in mice [114]. Moreover, metformin was shown to lead to an increased lifespan in nematode and rodent models [115,116]. Interestingly, some studies showed that the beneficial effects of metformin were less pronounced under autophagy-inhibited conditions, suggesting the importance of autophagy signaling induced by metformin [117,118,119,120]. It is now clear that metformin shows its beneficial effects at least partly through the induction of autophagy. However, in some models of aging, the longevity benefit of metformin was not observed. It is clear that metformin has several beneficial effects in various metabolic diseases. However, further investigation is needed to verify whether metformin can act as a CR mimetic and consistently present anti-aging effects.

### 5.3. Spermidine

Unlike rapamycin and metformin, spermidine is a natural polyamine that stimulates autophagy [121]. It has been demonstrated to be involved in various cellular processes and to regulate cellular homeostasis. The external supplementation of spermidine extends the lifespan in various species including yeast, nematodes, fruit flies, and mice [121,122,123]. It also showed protective effects in several degenerative diseases. Importantly, many of these anti-aging and beneficial properties of spermidine were abrogated when there was a genetic impairment to autophagy [123,124,125]. Mechanistic studies revealed that spermidine induces autophagy through the inhibition of several acetyltransferases. EP300, one of the acetyltransferases regulated by spermidine, is a main negative regulator of autophagy [126]. Epidemiology data showed that spermidine levels decline with age, and that the increased uptake of spermidine-rich foods diminishes the overall mortality associated with cardiovascular diseases and cancer [127,128]. Interestingly, a recent report also demonstrated a similar role for aspirin, and the induction of autophagy by aspirin has been demonstrated in several species [129,130]. Collectively, these results provide new molecular mechanisms for regulating autophagy, and spermidine and aspirin could form a new type of CR mimetics with anti-aging effects.

## 6. Concluding Remarks

In this review, the anti-aging effects of CR-induced autophagy were discussed. Although dependent on the species and age used in the experimental models and on the duration and intensity of CR regimens, all evidence supports a role for CR in autophagy activation. CR-induced autophagy plays a pivotal role under physiological conditions by maintaining adequate homeostasis in the organism. Furthermore, in various organs and tissues under pathologic conditions including aging, CR-induced autophagy played a protective role. The underlying mechanisms of longevity extension in response to CR are not yet fully understood, but evidence supports that activated autophagy could be playing an important role. With further advances in mechanistic biology, it is interesting that autophagy-inducing CR mimetics show similar effects to CR in several organisms. While more studies are required to better understand the benefits of CR mimetics, its safety and side effects should also be carefully considered. Finally, it will be necessary to assess whether autophagy inducers are effective and can be applicable in the treatment of human diseases.

## Figures and Tables

**Figure 1 nutrients-11-02923-f001:**
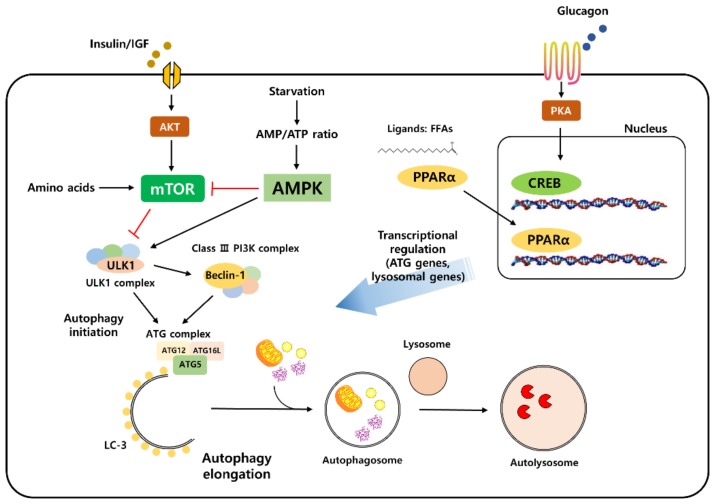
Autophagy is regulated by nutrient-sensing signaling. Autophagy signaling is modulated mainly by nutrient-sensing signaling pathways. Insulin and IGF (insulin-like growth factor) induce the activation of mammalian target of rapamycin (mTOR) signaling and inhibit autophagy initiation. The activation of AMP-activated protein kinase (AMPK) by an increased AMP/ATP ratio during starvation directly increases autophagy and inhibits the mTOR complex. CRE-binding protein (CREB) activation by glucagon signaling and peroxisome proliferation factor-activated receptor α (PPARα) activation by its ligands increases the gene transcription level of autophagy and lysosome-related proteins.

**Figure 2 nutrients-11-02923-f002:**
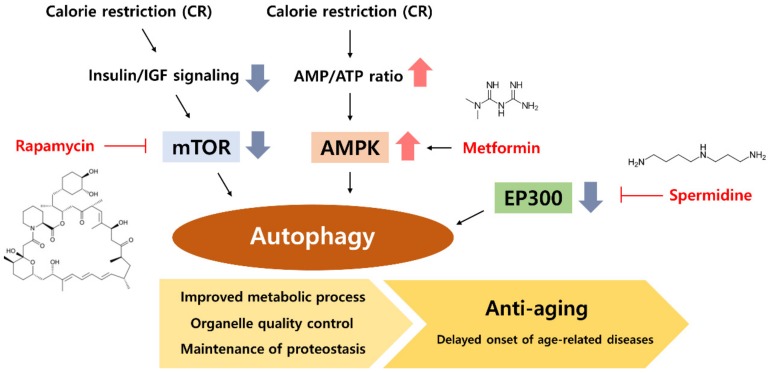
Calorie restriction (CR) and CR mimetics modulate the autophagy process. CR decreases mTOR signaling by reducing insulin and IGF levels. CR increases the AMP/ATP ratio and activates AMPK. Decreased mTOR and activated AMPK efficiently induce the initiation of the autophagy process. Various CR mimetics can induce the autophagy process. Rapamycin activates autophagy by inhibiting mTOR and metformin induces autophagy by activating AMPK. Spermidine enhances the overall autophagy process through the inhibition of EP300 deacetylase.

**Table 1 nutrients-11-02923-t001:** Studies showing protective effects of calorie restriction (CR)-induced autophagy in different organs. LC3: light chain 3.

Liver Autophagy			
Species	CR Methods	Main Results	Ref
Fisher Rats	40% calorie restriction (Life-long)	CR had no substantial effect on the expression of autophagic proteins	[62]
SD Rats	Alternative day fasting (10 months)	Alternative day fasting increased autophagy at high levels	[63]
Rat	40% calorie restriction (4 months)	CR increased autophagy flux (LC3-II/LC3-I ratio) especially in the mitochondrial membrane	[64]
Mice	0%–40% calorie restriction (4 months)	A significant increase in autophagy was detected	[65]
**Muscle Autophagy**			
Fisher Rats	8% calorie restriction (Life-long)	Mild CR attenuated the impairment of autophagy in rodent muscle during aging	[66]
Human	Up to 30% calorie restriction (3–15 years)	Autophagy-related genes were significantly increased in response to CR	[67]
Mice	40% calorie restriction (6–18 months)	Autophagy and mitochondrial integrity was significantly increased	[68]
**Adipose Tissue Autophagy**			
Mice	40% calorie restriction (15 days)	Autophagy was significantly induced in lean mice (but not in obese mice)	[69]
Mice	40% calorie restriction (Life-long)	Autophagy activity was enhanced in CR mice compare to aged mice	[70]
**Kidney Autophagy**			
Mice	40% calorie restriction (12 month)	Autophagy flux and LC3 conversion were higher in CR mice	[58]
SD Rats	40% calorie restriction (2 month)	Short-term CR increased LC3-II/LC3-I ratio and beclin-1 expression	[71]

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
