# Peer review of "The Effects of Calorie Restriction on Autophagy: Role on Aging Intervention"

_nutrients, 2019, doi:10.3390/nu11122923_

Round 1

Reviewer 1 Report

This paper is a review article on the effects of calorie restriction on autophagy and its role in aging intervention

Strengths: The authors do well describing the role of autophagy in maintaining cellular viability and homeostasis. They also describe the benefits of calorie restriction (CR), which activates autophagy, on lifespan extension and general well-being. Their description of autophagy in various organs is interesting, as CR appears to regulate autophagy in these organs to varying degrees. Finally, the authors point out the relative values of CR mimetic agents in autophagy activation. These include administration of or treatment with  rapamycin, polyamines, aspirin, and metformin as well as alternate strategies to CR, such as inter-meal fasting.

Weaknesses: Overall, the writing is good, but this reviewer suggests that the paper be proofed by a native English speaker. There were instances of lack of agreement between subject and verb (e.g. line 216). Some sentences had words missing (e.g. line 330  the article "a" should precede "role").  Line 375, "protein translation" should be replaced with "protein synthesis".

Organ  Distinction and tense: The authors should not use the terms "metabolic and non-metabolic organs". All organs are metabolic but some organs exhibit more prominent metabolic pathways (e.g. gluconeogenesis) than others. Authors should use present tense when describing published work.  

Clarity: CR, in most cases refers to intake of 60% of calories or a 40% reduction in calories. Authors need to  make that point clear, as it appears to differ in different parts of the paper. 

Author Response

Response to reviewer: Thanks for your insightful comments. We modified manuscripts accordingly.

Reviewer 2 Report

This review article deals with a possible effect of calorie restriction (CR) to aging organs (liver, muscle, adipose tissue, kidney) by activating autophagy. The authors wrote comprehensively and brilliantly this topic with referring many recent articles.

In mammalian skeletal muscle, autophagic defect in sarcopenia is accepted for recent 10 years. In the part of 1.4 Autophagy and aging, you should add the important sentences using following references.

Wenz T, Rossi SG, Rotundo RL, et al. (2009) Increased muscle PGC-1alpha expression protects from sarcopenia and metabolic disease during aging. Proc Natl Acad Sci U S A 106:20405-20410 Wohlgemuth SE, Seo AY, Marzetti E, et al. (2010) Skeletal muscle autophagy and apoptosis during aging: effects of calorie restriction and life-long exercise. Exp Gerontol 45:138-148 Carnio S, LoVerso F, Baraibar MA, et al. (2014) Autophagy impairment in muscle induces neuromuscular junction degeneration and precocious aging. Cell Rep 8:1509-1521 Sakuma K, Kinoshita M, Ito Y, et al. (2016) p62/SQSTM1 but not LC3 is accumulated in sarcopenic muscle of mice. J Cachexia Sarcopenia Muscle 7:204-212

Garcia-Prat L, Martinez-Vicente M, Perdiguero E, et al. (2016) Autophagy maintains stemness by preventing senescence. Nature 529:37-42  Jiao J, Demontis F (2017) Skeletal muscle autophagy and its role in sarcopenia and organismal aging. Curr Opin Pharmacol 34:1-6

Author Response

Response to reviewer: Thanks for your insightful comments. We added references on the section 3.2 about the defective autophagy in the sarcopenia.

Reviewer 3 Report

The manuscript attempts to provide an overview of findings highlighting caloric restriction as an important factor in regulation of autophagy as a lifespan extending pathway. The manuscript is poorly reads and unfocused between the role of CR on aging and autophagy, and the role of autophagy on aging. There are multiple unsettling and vague statements throughout the manuscript at times going into mere listing of findings in the literature

Several examples and my other comments are below:

Section1.4 line 129: “whole body deletion of autophagy-related genes led to early postnatal death indicating an essential role of autophagy in the overall maintenance of physiological processes” does not indicate or support a role for autophagy in aging but perhaps in development.

A review is expected to be clarifying and precise in representing the reported findings. The first paragraph in section 1.4 highlights reported decrease in overall autophagic capacity during aging and decreased expression of autophagy related genes, and continues on with findings that are showing accumulation of autophagic vacuoles apparently due to impairments in proteolytic capacity with aging creating confusion as to how accumulation of autophagic vacuoles fits with the decreased expression of autophagy related genes.

Section 1.4. The first paragraph in section 1.4 highlights observed decrease in overall autophagic capacity during aging and decreased expression of autophagy related genes and continues on (line 123) with findings that are showing accumulation of autophagic vacuoles apparently due to impairments in proteolytic capacity. How these findings go together and how accumulation of autophagic vacuoles fits with decreased expression of autophagy related genes.

The title for Table 1 is misleading. Table shows studies analyzing the CR effect on autophagy in different organs. The title “Protective effect of CR-induced Autophagy” is not a correct description of what is presented in the table.

Line 180: “CR showed a prolonged lifespan in a wide range of species from yeast to non-human primates, and in humans as well” should be corrected to read as “CR showed a prolonged lifespan in a wide range of species from yeast to non-human primates, and supports a healthy human aging”.

References should be incorporated in the figures1 and 2 or in the figure legends.

Manuscript contains multiple grammatical errors. Examples:

Line 29: Change the word evolutionary to 'evolutionarily"

Line 82: Should read as "presence of nutrients"

Author Response

The manuscript attempts to provide an overview of findings highlighting caloric restriction as an important factor in regulation of autophagy as a lifespan extending pathway. The manuscript is poorly reads and unfocused between the role of CR on aging and autophagy, and the role of autophagy on aging. There are multiple unsettling and vague statements throughout the manuscript at times going into mere listing of findings in the literature

Response to reviewer: Thanks for your insightful comments. We tried to modify and clarify our manuscripts according to comments.

Several examples and my other comments are below:

Section1.4 line 129: “whole body deletion of autophagy-related genes led to early postnatal death indicating an essential role of autophagy in the overall maintenance of physiological processes” does not indicate or support a role for autophagy in aging but perhaps in development.

Response to reviewer: Thanks for your comment. I agree with the reviewer’s suggestion. However, the sentence was added just to inform the reader that the whole body knockout mice are not available due to postnatal death and to emphasize the physiological role of autophagy.

A review is expected to be clarifying and precise in representing the reported findings. The first paragraph in section 1.4 highlights reported decrease in overall autophagic capacity during aging and decreased expression of autophagy related genes, and continues on with findings that are showing accumulation of autophagic vacuoles apparently due to impairments in proteolytic capacity with aging creating confusion as to how accumulation of autophagic vacuoles fits with the decreased expression of autophagy related genes.

Response to reviewer: Thanks for your comment. As reviewer’s comments, it can be confusing since autophagic vacuoles usually increases with autophagy process. However, it also means overall autophagy process is stopped due to decrease in autophagy flux. We changed the sentence to avoid the confusion.

Section 1.4. The first paragraph in section 1.4 highlights observed decrease in overall autophagic capacity during aging and decreased expression of autophagy related genes and continues on (line 123) with findings that are showing accumulation of autophagic vacuoles apparently due to impairments in proteolytic capacity. How these findings go together and how accumulation of autophagic vacuoles fits with decreased expression of autophagy related genes.

Response to reviewer: Thanks for your comment. The sentence is replaced to avoid the confusion. For the proteolytic capacity, we tried to explain over all protein degradation is decreased during aging. Because protein can be degraded in two way, ubiquitin-proteasome and autophagic degradation, we meant that whole proteolysis is decreased during aging.

The title for Table 1 is misleading. Table shows studies analyzing the CR effect on autophagy in different organs. The title “Protective effect of CR-induced Autophagy” is not a correct description of what is presented in the table.

Response to reviewer: Thanks for your comment. We agree with reviewer’s comment. We changed the title.

Line 180: “CR showed a prolonged lifespan in a wide range of species from yeast to non-human primates, and in humans as well” should be corrected to read as “CR showed a prolonged lifespan in a wide range of species from yeast to non-human primates, and supports a healthy human aging”.

Response to reviewer: Thanks for your comment. We agree with reviewer’s comment. We changed the sentence.

References should be incorporated in the figures1 and 2 or in the figure legends.

Manuscript contains multiple grammatical errors. Examples:

Line 29: Change the word evolutionary to 'evolutionarily"

Line 82: Should read as "presence of nutrients"

Response to reviewer: Thanks for your comment. We changed grammatical errors.